# High reliability organizations and healthcare safety outcomes on patients and staff: Scoping review

Michael Joshua G. Morales[1,2]*, Pauline Hilton[1], OiSaeng Hong[1], Stella Bialous[1], Marie Martin[2], Jessica Sewnath[2], Soo-Jeong Lee[1]

**1** University of California, San Francisco, California, United States of America, **2** Department of Veterans Affairs, Washington DC, United States of America

* Michaeljoshua.morales@ucsf.edu, michaeljoshua.morales@va.gov

## Abstract

Adverse events such as medication errors and staff injuries are prevalent in healthcare and contribute to patient harm and staff burnout. To enhance safety, organizations implemented the high-reliability organization (HRO) principles which aim to maintain low rates of adverse events while managing complex processes. These principles include sensitivity to operations, preoccupation with failure, reluctance to simplify, resilience, and deference to expertise. To explore evidence on HRO implementation and its impact patient and staff safety outcomes in healthcare. A scoping review of English-language studies published from 2016 to 2025. A librarian was consulted to develop electronic search strategies. Three databases were utilized to identify the relevant studies. Inclusion criteria were studies on HRO implementation in healthcare and safety outcomes on patient or staff. Two reviewers independently screened titles and abstracts, assessed full texts, and extracted data using the Joanna Briggs Institute (JBI) data extraction tool, with a third reviewer available to resolve disagreements. Of 3,305 studies retrieved, eleven met inclusion criteria. The studies focused on patient safety (e.g., medication errors, falls; n = 6). Two explored staff perceptions of patient safety; two assessed both staff perceptions and patient outcomes. One addressed staff safety. HRO implementation was associated with fewer adverse events, positive staff perceptions on patient safety, and improved psychological safety. Evidence suggests that HRO implementation is positively associated with improved patient safety and staff psychological safety. However, evidence on their impact on staff physical safety remains limited.

Health care errors can result in serious harm to patients and staff, including physical injuries and impairment, psychological distress, and even death [1,2]. Annually, healthcare-related errors result in approximately 400,000 patient injuries and 100,000 deaths in the United States [3]. In addition, the healthcare industry's worker injury rate was 4.5 cases per 100 full-time equivalents (FTE) in 2022, which was

**Data availability statement:** This study is a scoping review and did not generate or analyze any new datasets involving human participants. All relevant data underlying this review are contained within the published articles included in the analysis, which are fully listed in Table 3 of the manuscript. No additional datasets were used.

**Funding:** This study was financially supported by the National Institute for Occupational Safety and Health in the form of a grant awarded to MM (3T42 OH008429). No additional external funding was received for this study. The funders had no role in study design, data collection and analysis, decision to publish, or preparation of the manuscript.

**Competing interests:** The authors have declared that no competing interests exist.

substantially higher than the rates of the construction industry (2.4 per 100 FTE) or the agricultural, forestry, fishing, and hunting industry (4.1 per 100 FTE) [4]. Legislation and regulations have led to organizational programs aimed to improve patient and staff safety and mitigate healthcare-related errors and work-related injuries; however, adherence to legislation and regulations alone is not sufficient to substantially reduce the frequency and severity of such incidents [1,5].

High-reliability organizations (HROs) are institutions that conduct complex operations with the goal of zero errors or adverse outcomes, and the adoption of HRO concepts and principles can be a promising approach in the healthcare system for patient and staff safety [6–9]. The healthcare industry involves a complex multisystem that can be susceptible to errors that may lead to catastrophic outcomes, such as patient death [1,10]. The concept of HRO in healthcare was initially introduced in 1998 by the Institute of Medicine (IOM) and was formally supported by the Agency of Healthcare Research and Quality in 2005 through HRO implementation guidelines for healthcare institutions [1,11]. The aim of HRO implementation in healthcare was primarily to reduce healthcare-related errors and injuries for patients, but its impact may also apply to staff safety. HRO fosters a culture of safety by promoting a culture wherein the blaming of errors is discouraged, building staff trust with leadership, learning from the organization's errors, and adapting to a rapidly changing operational situation [12–14]. Organizational efforts to improve the culture of safety are regarded as a key component in promoting cultural and behavioral changes [15].

HROs are guided by five fundamental principles: preoccupation with failure," "reluctance to simplify," "sensitivity to operations," "commitment to resilience," and "deference to expertise" [14]. The principle of *pre-occupation with failure* views near-missed errors as learning opportunities to increase caution owing to the absence of errors within the operation [16,17]. The principle of *reluctance to simplify* considers the complexity of the source of failure and recommends avoiding oversimplified explanations of problems [16,17]. The principle of *sensitivity to operations* refers to an organization's heightened vigilance regarding potential threats or sources of errors within their systems or processes [16,17]. The principle of *commitment to resilience* refers to a focus on continuous error correction to minimize the negative impact of errors on the organization [16,17]. The principle of *deference to expertise* refers to prioritizing the knowledge and input of safety from the most informed individuals, regardless of rank or position [16,17].

Leadership is crucial in organizational implementation. Chassin and Loeb [17] discussed the indispensable role of leadership commitment to zero harm in achieving sustainable high reliability and underlined the crucial role of senior leadership in providing continuous support to front-line leaders. Implementing HRO involves the evaluation of progress; a guiding theoretical model is helpful [7]. One model was developed by the Joint Commission's High Reliability Healthcare Maturity model (HRHCM) and classified an organization's HRO level as *beginning, developing, advancing,* and *approaching* HRO [17,18]. The HRHCM model defines the maturity level based on the organization's leadership, culture of safety, and robust process improvement [17,18]. Evaluation tools, such as Randall et al.'s [19] HRO HRHCM

Questionnaire and Oro 2.0, by the Joint Commission [7] were developed to determine the HRO maturity levels. The Institute for Healthcare Improvement (IHI) Framework for Safe, Reliable and Effective Care serves as a guiding HRO model by instilling the core values of psychological safety, accountability, teamwork, communication, and negotiation [7].

Veazie et al. [7] conducted a rapid evidence review in preparation for the nationwide implementation of HRO in the Department of Veteran Affairs (VA) beginning in February 2019. The review included twenty-six studies from 2010 to 2019; eight studies were on HRO implementation frameworks, nine studies were on HRO evaluation metrics, and nine studies focused on the impact of HRO in the healthcare system. Studies on the impact of HRO in healthcare have shown that HRO implementation decreases serious patient harm, such as medication errors and hospital-acquired infections [7]. However, no study examined the impact of HRO implementation on staff safety. Thus, the purpose of this scoping review was to systematically identify and synthesize more recent research literature on HRO with respect to both patient and staff outcomes.

## Method

The protocol used in this scoping review included key sections that aligned with the Joanna Briggs Institute (JBI) guidelines (S1 Checklist): the introduction, search strategies, inclusion and exclusion criteria, study selection, data extraction and analysis, and results.

### Ethical statement

This study was conducted as a scoping review and did not involve human or animal subjects; therefore, no ethical approval was required. All the data analyzed in this review were obtained from publicly available sources, adhering to the principles of research integrity and transparency.

### Search strategies

This study used three databases, PubMed, Web of Science, and Embase. The search terms were based on previous HRO evidence reviews and theoretical papers. To ensure that pertinent studies were not omitted, a research librarian developed a comprehensive search strategy for each database and obtained search terms from the titles and abstracts of relevant articles, as well as from the index terms used to describe the articles (see Table 1). The search scope was limited to studies published in English between January 2016 and September 2025. This time period was selected to incorporate relevant studies beyond 2019, extending prior reviews and capturing more recent evidence on the application and outcomes of HRO principles in healthcare.

### Inclusion and exclusion criteria

Eligible articles were studies on healthcare populations: healthcare staff, patients, and healthcare organizational entities. The included studies were empirical studies that used various research methods such as qualitative inquiry, cross-sectional, quasi-experimental, and retrospective designs, as well as mixed methods approaches. Articles that explicitly investigated the implementation of HRO, with a focus on promoting safety for patients and staff, were considered for inclusion in the analysis.

### Study selection

Following the search, the citations of the selected studies were collated and uploaded into Zotero version 6.0.26. Potentially relevant sources were retrieved in full text, and their citation details were imported into Covidence, an evidence review software. The search results and study inclusion process are reported in the Preferred Reporting Items for Systematic Reviews and Meta-analyses extension for scoping review (PRISMA-ScR) flow diagram [20].

**Table 1. Search Terms.**

| Full Search Strategy with Librarian consultation (May 5, 2023) | | | |
|---|---|---|---|
| **Database** | **Limitations Applied** | **Search Terms** | **Results** |
| PubMed | 2016-2025 English | (HRO OR "high-reliability organization" OR "high reliable organization" OR "high reliability teams" OR ("Organizations"[Mesh] AND (reliable OR reliability))) AND (safety OR "Safety"[Mesh] OR "safety outcomes" OR "safe outcomes" OR "culture of safety" OR "Safety Management"[Mesh] OR "safety management" OR "Quality Improvement"[Mesh] OR "quality improvement" OR "Outcome Assessment, Healthcare"[Mesh] OR "outcome assessment") | 449 |
| Web of Science | 2016-2025 English Articles | (HRO OR "high-reliability organization" OR "high reliable organization" OR "high reliability teams" OR ("Organizations" AND (reliable OR reliability))) AND (safety OR "safety outcomes" OR "safe outcomes" OR "culture of safety" OR "safety management" OR "quality improvement" OR "outcome assessment") | 1,812 |
| Embase | 2016-2025 English Articles/Articles in Press | (hro OR 'high-reliability organization'/exp OR 'high-reliability organization' OR 'high reliable organization' OR 'high reliability teams' OR (('organizations'/exp OR 'organizations') AND (reliable OR 'reliability'/exp OR reliability))) AND ('safety'/exp OR safety OR 'safety outcomes' OR 'safe outcomes' OR 'culture of safety' OR 'safety management'/exp OR 'safety management' OR 'quality improvement'/exp OR 'quality improvement' OR 'outcome assessment'/exp OR 'outcome assessment') | 1,044 |
| Total | | | 3,305 |

Two independent reviewers (MM and JS) with experience in healthcare HRO implementation conducted the inclusion-screening process. The reviewers screened the titles and abstracts and then conducted detailed assessments of the full texts of the selected articles. Any disagreements between the reviewers were resolved through discussions at each stage of the selection process. An additional reviewer was on standby to serve as a tiebreaker in case resolution could not be achieved through discussion but was not needed because two reviewers reached a consensus for all cases of disagreement after discussions.

A combined search of PubMed, Embase, and Web of Science identified 3,305 articles. Among these articles, 524 were duplicates, leaving 1,960 for title and abstract screening. A total of 1,853 articles were excluded, primarily because of the term "reliability" in the search criteria encompassing studies on instrument reliability and validity. Consequently, 107 studies were eligible for full-text review, with three full-text studies being unattainable because of unavailability through the university library system. Upon full-text review, 93 studies were excluded, predominantly because 58 articles were evidence-based projects, evidence synthesis reviews, conference presentations, theoretical framework papers, and book chapters. Moreover, 19 articles were excluded because they did not discuss the organizational application or alignment of HRO principles or because they discussed the concept of HRO as an outcome rather than as a predictor variable. Therefore, a total of 11 studies were included in this review. The selection process is presented in Fig 1, along with the reasons for exclusion.

## Data extraction and analysis

Using the JBI data extraction tool (Table 2), data from the selected studies were extracted by the same two reviewers who screened and selected the pertinent studies. The extracted data included the authors, aims of the study, study participants, sample size, study methods, instrument tools used, outcome measures, and key findings relevant to the review purpose. As in the article inclusion process, any disagreement between the reviewers was resolved through discussion to reach a final decision. A third reviewer was available to adjudicate unresolved disagreements; however, third-party arbitration was not required. A quality assessment of the selected studies was not performed because scoping reviews do not have formal quality assessment methods [21–23]. The data analysis included a narrative and qualitative examination of study characteristics, and an exploration aimed at identifying themes in the synthesized studies.

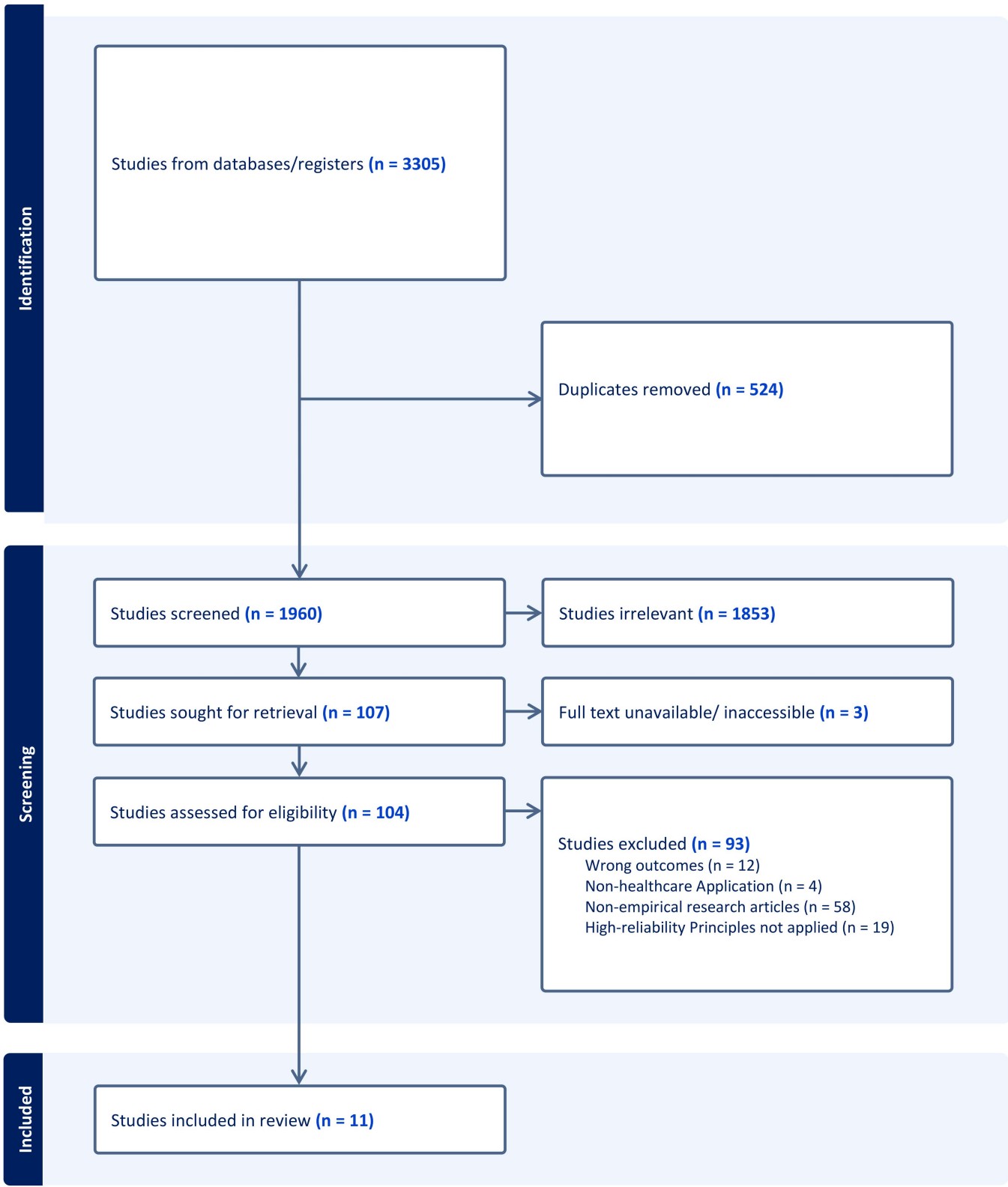

**Fig 1. Flow diagram depicting the identification, screening, eligibility, and inclusion of studies for this scoping review, adapted from the PRISMA Extension for Scoping Reviews (PRISMA-ScR) guidelines.**

**Table 2. JBI Data Extraction Tool.**

| Sections | Sub-sections |
|---|---|
| Citation Details | Title |
| | Authors |
| | Years of Publication |
| | Country of Origins |
| | Aims or Purpose |
| Population and Characteristics | Population Characteristics |
| | Sample Size |
| Methods | Type of Study<br>Quantitative Study<br>1. Cross-sectional<br>2. Prospective<br>3. Case-control<br>4. Retrospective<br>5. Randomized Control Trial<br>Qualitative/Mixed Methods<br>1. Qualitative<br>2. Mixed-Methods |
| | Methods |
| | Instrument Tool Used |
| | Dates of Data Collection |
| Outcomes | Outcomes on Patient or Staff<br>1. Patient<br>2. Staff<br>3. Others |
| | Outcome Measures |
| | Key Findings |

## Results

### Years of publication and country

Ten studies included in this review were published between 2016 and 2025, with no studies published in 2020 or 2024 (Table 3). Geographically, eight studies were conducted in the U.S., indicating a concentrated application of HRO in this region. The remaining two studies were conducted in Canada and New Zealand, respectively.

### Types of study

The majority (73%) of the selected studies used quantitative methodologies, including five cross-sectional designs, two prospective design, and one pre-post quasi-experimental design. One study employed a qualitative design. Two studies used a mixed-method approach.

### Characteristics of samples

The studies in this review used diverse samples consisting of either organizational entities or healthcare staff. The study settings included in the studies ranged from one to 139 healthcare facilities, and staff participants were recruited from various departments such as medical-surgical units, procedural areas such as the catheterization lab, pediatric units, intensive care unit, and the emergency department [9,13,24–30]. Randall et al. [19] and Lyren et al. [31] focused on hospital organizations as entities and recruited hospitals from the same children's hospital network.

**Table 3.** *Included Studies.*

| First Author. Year, Country | Independent Variable | Dependent Variable | Sample | Type of Study | Instrument used | Key Findings |
|---|---|---|---|---|---|---|
| **Patient Safety Perception** | | | | | | |
| Goldstein et al. 2017 Canada | HRO Principles | Patient Safety Climate (perception) | Hospital Leadership- 136 respondents a.Board/ Administration- 28 b. Medical/ Nursing Leadership-108 | Cross-sectional | The Patient Safety Culture in Health-care Organization (PSCHO) survey | • Perceptions of safety climate differed significantly between board/administrative leaders and medical/nursing leaders. Medical/nursing leadership more like to respond problematic or neutral safety climate perception. • Leadership had a poor alignment with their perception of safety and HRO principles. |
| Wailling et al. 2018 New Zealand | Implementation of HRO in the organization | Patient safety perception | Doctors- 31 Registered Nurses- 19 Senior Leadership- 3 | Qualitative | Semi-structured interview | The key themes identified with the study includes: • Participants' comprehension of patient safety: • Safety occurs in complex systems • Safety surveillance • Vigilance and anticipation • Risk management strategy • Alignment with HRO characteristics/principles: • Safety surveillance: mindfulness, resilience • Vigilance/anticipation: deference to expertise |
| **Patient Safety Outcomes** | | | | | | |
| Vogus and Iacobucci 2016 United States of America | Reliability Enhancing Work-Practices | Medication errors and patient falls. | RNs- 1,685 | Cross-sectional | Reliability Enhancing Work Practice Survey Secondary analysis on patient falls and medication errors. | • An increase in REWPs was associated with a decrease in medication errors (random effects -0.11, $p < 0.05$) and patient falls (random effects -0.39, $p < 0.001$). |
| Lyren et al. 2017 United States of America | Implementation and training of HRO principles | Hospital acquired conditions (HAC) and serious safety events (SSE) | Children's Hospital-32 | Prospective- 3 years (2012–2015) | Secondary analysis of HAC and SSE | • Outcomes of the improvement collaborative program on 8 out of 9 common HACs: reduction from 9% to 71% ($p < 0.005$); venous thromboembolism exhibited no significant change. • SSE rate: 32% reduction, from 0.77 to 0.52 per 10,000 patient days of care ($p < 0.001$) |
| **First Author. Year, Country** | **Independent Variable** | **Dependent Variable** | **Sample** | **Type of Study** | **Instrument used** | **Key Findings** |
| Roney et al. 2017 United States of America | HRO safety culture training | Patient safety events like medication errors, and near misses. | Baccalaureate Nursing faculty- 18 | Mixed Methods | Association of Healthcare Research and Quality (AHRQ) Focus group questionnaire | • The following themes were identified: • HRO training bolstered awareness of clinical practice deviations, fostering a 'solve-it' rather than 'fix-it' approach to root causes. • It enhanced faculty's safety mindset, empowering them to comfortably adopt HRO processes, like effective communication and safety interventions. • The value of reporting near misses and safety events was recognized as a tool for safety improvement. |
| Mossburg et al. 2019 United States of America | Patient safety profiles | Manifestation of HRO principles in staff language. | Total Nurses=41 High Performing Unit Nurses- 20 Low Performing Unit Nurses- 21 | Mixed Methods | Safety Attitudes Questionnaire and Hand Hygiene Adherence Semi-structured interview | • Higher performing units with elevated safety profiles displayed higher manifestation of HRO principles in staff's work language. |

*(Continued)*

**Table 3.** (Continued)

| First Author. Year, Country | Independent Variable | Dependent Variable | Sample | Type of Study | Instrument used | Key Findings |
|---|---|---|---|---|---|---|
| Randall et al. 2021 United States of America | HRO maturity level | HAC and serious harm events (SHE) | Children's Hospital- 25 | Cross-sectional | High Reliability Healthcare Model Questionnaire Secondary analysis of HAC from database to SHE. | • No significant association between HRO level and SHE index (OR=0.91, 95% CI: 0.78-1.05)<br>• Significant relationship between culture of safety domain in HRO and SHE index (OR=0.63, 95% CI: 0.42-0.95) |
| Pfeifer et al. 2023 | Working in a HRO facility | Staff Psychological Safety Patient Safety Reporting Intention | 224 Pediatric Nurses | Cross-sectional | Safety Organizing Scale (HRO measure) Intention to Report Safety Events Scale | • Nurses working in facilities perceived to have a higher HRO status where more likely to report safety events (p<.05). |

**Patient Safety and Perceptions of Patient Safety**

| First Author. Year, Country | Independent Variable | Dependent Variable | Sample | Type of Study | Instrument used | Key Findings |
|---|---|---|---|---|---|---|
| Sculli et al. 2022 United States of America | Implementation and training of HRO principles | Patient Safety Culture Patient Safety event reporting | VA Hospital-1 Frontline Staff Respondents: 2016- 232 2019- 1138 | Prospective-3 years (January 1, 2016–December 31, 2018) | Patient Safety Culture Survey Secondary analysis of Patient Safety Reports | • Reporting of non-serious safety events increased significantly (p<.001), rising from 241 per 10,000 patients pre-intervention to 382.1 per 10,000 patients post-intervention.<br>• Potential serious safety events decreased significantly (p<.001), dropping from 103.9 per 10,000 patients to 41.9 per 10,000 patients post-intervention.<br>• No change in serious safety reporting (p=.168)<br>• Improved patient safety culture compared to the national VA average (p<.001) |

| First Author. Year, Country | Independent Variable | Dependent Variable | Sample | Type of Study | Instrument used | Key Findings |
|---|---|---|---|---|---|---|
| Sawyer et al. 2025 United States of America | Implementation of HRO in an organization | Patient Safety Culture | Total VHA Facilities-139 Intervention group Cohort 1–18 Comparison group-121 | Pre-Post Quasi-Experimental | All- Employee Survey (Patient Safety Culture Domain) Secondary data analysis of patient safety events reporting, close calls, adverse events (AE), and serious safety events (SSE). | • Cohort 1 showed greater gains during intervention and post-intervention on the following Patient Safety Culture domains: Teamwork (+4.87 vs 4.41, p=.04), just culture (+0.32 vs -0.60, p=.004), error transparency (+4.95 vs 3.25, p<.001).<br>• Cohort 1 had a higher patient safety event reporting compared to the comparison group during the intervention year and post-intervention year 1 (p<.05).<br>• Cohort 1 showed a slight increase in AE and the control group with a decrease (p=.025). There was no significant difference on SSE between the groups.<br>• HRO implementation showed an improvement in most patient safety culture domains. Patient safety reporting was higher in facilities that implemented HRO. |

**Staff Outcomes**

| First Author. Year, Country | Independent Variable | Dependent Variable | Sample | Type of Study | Instrument used | Key Findings |
|---|---|---|---|---|---|---|
| Gilmartin et al. 2022 United States of America | Learning Environment and High-Reliability Practice (LEHR) | Employee engagement, employee retention, and safety climate. | Total Personnel: 232 Nurses: 146 Interventional Cardiologists: 27 Technicians: 42 Supervisory roles: 58 Other: 17 | Cross-sectional | Learning Environment and High Reliability Practice (LEHR) Demographic Questionnaires Employee Engagement, retention, safety climate. | • There is a significant relationship between LEHR scores and employee engagement, retention and safety climate. Higher LEHR scores associated with: Higher job satisfaction (chi-square 260.06, p<0.001)• Lower burnout (chi-square 49.2, p<0.001)<br>• Lower intent to leave (chi-square 85.4, p<0.001)<br>• Higher perceived safety climate (chi-square 156.8, p<0.001)<br>• Lower turnover rate in 12 months (chi-square 23.9, p<0.001) |

For staff samples, Pfeiffer et al. [26], Vogus and Iacobucci [29], and Mossburg et al. [13] included solely licensed nurses (i.e., the studies excluded non-licensed nursing staff, such as nursing assistants). Gilmartin et al. [24], Wailling et al. [30], and Goldstein et al. [25] included diverse occupations, including nursing staff, medical staff, allied health staff, non-licensed personnel (e.g., healthcare technicians), and leadership teams.

## HRO measures

A range of measures has been employed to assess the implementation and application of HRO, reflecting the diverse approaches used to capture HRO maturation and integration within organizations. Randall et al. [19] assessed HRO implementation using a distinct measure of the HRHCMM questionnaire, which focuses on the inherent levels of HRO maturation. Several other studies incorporated additional concepts to measure HRO implementation. For example, Gilmartin et al. [24], Vogus and Iacobucci [29], Pfeiffer et al. [26], and Goldstein et al. [25] quantitatively measured the degree of HRO principle implementation.

Roney et al. [27], Mossburg et al. [13], and Wailling et al. [30] qualitatively measured the presence of HRO. The studies used semi-structured interviews to identify the HRO languages and principles embedded in participants' narratives. Sculli et al. [9], Sawyer et al. [28], and Lyren et al. [31] correlated HRO with differences in patient safety outcomes by comparing data from pre and post implementation of HRO programs.

## Patient safety outcomes

Five of the eight studies focused on *patient safety events* events that elevate the risks of hazards to patients' health or safety. Variables measuring patient safety events included medication errors, patient falls, hospital-acquired conditions, safety unit profiles, and serious safety events (SSE) —measured as the total number of hospital-acquired infections, patient falls, medication errors, and surgical safety events in an organization.

Randall et al. [19] examined the impact of the HRO maturity level on patient safety events, including hospital-acquired conditions, serious safety events, and serious harm index. Randall et al. [19] found no significant relationship between the overall HRO maturity level and the serious harm index ($OR = 0.91$, 95% CI [0.78, 1.05]). However, the culture of safety domain of the HRHCM questionnaire was associated with decreased serious harm index (SHI) ($OR = 0.63$, 95% CI [0.42, 0.95]). In a study by Lyren et al. [31], the application of HRO principles was associated with lower levels of hospital-acquired conditions namely: adverse drug events, catheter associated infections, central line associated bloodstream infections, falls, obstetrics adverse events, pressure ulcers, surgical site infections, and ventilator associated pneumonia (reduction of 71%; $p < .005$); and serious safety events (reduction of 32%; $p < .001$).

Roney et al. [27] and Vogus and Iacobucci [29] explored the relationship between HRO and patient-safety events pertaining to medication errors. Roney et al.'s [27] qualitative study findings revealed that HRO training improved the awareness of clinical practice deviation, finding the root cause of medication issues, empowerment to adopt HRO principles and mindsets, and the value of reporting safety events. Vogus and Iacobucci [29] found that reliability-enhancing work practices were associated with decreased medication errors ($\beta = -0.11$, $p < .05$) and patient falls ($\beta = -0.39$, $p < .01$). Lastly, Pfiefer et al. [26] found that staff working in an HRO facility were more likely to report patient safety events compared to non-HRO facilities ($p < .05$).

Mossburg et al. [13] examined the overall safety climate by compliance with hand hygiene practices, ranking the patient safety profiles of individual units to determine the high and low safety profiles. The qualitative outcome showed that more HRO-related language like "mindfulness" and principles were found in units with higher safety profiles in each transcription.

Roney et al.'s [27] and Mossburg et al.'s [13] mixed-method studies showed that HRO principles were embodied in narrative statements that emphasized the importance of routine practices and experiences pertinent to patient safety event management. Randall et al. [19] found that only the culture of safety component of the HRHCMM was significantly associated with reduced serious harm index, underscoring the critical role of safety culture in the HRO principles in improving patient safety outcomes.

## Staff's perceptions of patient safety

Two studies addressed patient safety perceptions from the perspective of healthcare staff [25,30]. Goldstein et al. [25] focused on examining the present state of patient safety within their specific organization, whereas Wailling et al. [30] viewed patient safety as a broader conceptual framework. In Goldstein et al.'s [25] study, more than 14 of the overall organizational leadership perception of patient safety were classified as problematic or neutral indicating misalignment with HRO principles in the organization. Medical and nursing leadership were more likely to report problematic or neutral responses than executive board leadership.

Wailling et al. [30] conducted semi-structured interviews to explore staff members' perceptions of patient safety in relation to the principles and characteristics of HRO. The themes that emerged from the study included the following: (a) staff viewed patient safety as present in complex systems, (b) identified safety surveillance as a critical component for determining system deficiencies and attenuating risk, (c) posited that vigilance and anticipation involved adaptability and active response to patient safety hazards as opposed to evasion, and (d) advocated for the implementation of risk management strategies. Among these themes, safety surveillance and vigilance anticipation are related to the characteristics and principles of HRO. Wailing et al. [30], described safety surveillance corresponding to the concept of mindfulness, where past failures are not only acknowledged but are also applied as learning opportunities. Vigilance and anticipation are linked to the principle of deference to expertise. Additionally, Wailing et al. [30] found a significant association between staff perceptions of patient safety and HRO principles, with safety being recognized as inherent to complex operations.

## Patient safety outcomes and staff perceptions of patient safety

Two studies both examined patient safety outcomes and staff perception on patient which similarly used a quasi-experimental design and conducted the studies in Veteran Affairs (VA) health care facilities. First, Sculli et al. [9] examined patient safety events and staff perceptions of patient safety before and after HRO implementation at the Truman VA Medical Center from 2016–2018. The study classified patient safety events into nonserious or non-fatal harm without permanent loss of function, potential serious or near misses that could have caused severe harm but were prevented, and serious or events resulting in death or permanent injury. Reporting of non-serious events increased significantly after HRO implementation (p<.001), while potential serious events decreased significantly (p<.001), with no significant change in actual serious events (p=.0168). Additionally, the study demonstrated significant improvements in patient safety culture among frontline staff, surpassing VA system-wide averages (p<.001).

Second, Sawyer et al. [28] compared the patient safety culture and patient safety event outcomes (event reporting, close calls, adverse events or events that caused harm but not permanent bodily harm or death, and serious safety events or event that led to death or permanent bodily harm) in a pre-post HRO implementation study across 18 VA facilities or Cohort 1, with 121 VA facilities serving as the comparison group. Patient safety culture data were obtained via an agency-wide individual employee survey aggregated at the facility level, and patient safety event outcomes were sourced from existing each VA's safety event reports. The results showed that Cohort 1 compared to the comparison group had greater gains either during the HRO implementation or post-implementation in several patient safety culture domains, teamwork (+4.87 vs. +4.41, p=.04), just culture (+0.32 vs. -0.60, p=.004), and error transparency (+4.95 vs. +3.25, p<.001). The trust domain did not show significant between-group differences during HRO implementation or post-implementation (p>.05). Reporting of safety events where higher in Cohort 1 during the HRO implementation and one-year post-implementation (p<.05). Adverse events showed a slight increase in Cohort 1 while decreasing in the comparison group, resulting in a significant overall between-group difference (p=.025). There was no significant difference between the two group when it comes to serious safety events.

## Staff outcomes

Among the reviewed studies, only Gilmartin et al. [24] examined the impact of HRO and working environment on staff outcomes such as overall engagement, retention, and work safety climate among cardiac catheterization unit staff. The

study showed that higher high reliability practices (scored from 1-7) was associated with higher job satisfaction ($p < .001$), lower burnout ($p < .001$), lower intent to leave ($p < .001$), perceived safer working climate ($p < .001$), and a lower turnover rate ($p < .001$).

## Discussion

This scoping review was conducted to identify and synthesize evidence pertaining to HRO application in healthcare organizations and its effects on patient safety and staff outcomes. This review included eleven studies from 2016 to 2025. The overall findings suggest that HRO principles application improve patient safety and staff outcomes and promote positive perceptions about patient safety among staff.

Despite the seemingly small number of included studies, this is not uncommon for scoping reviews and rapid evidence reviews with strict inclusion criteria focused on empirical research. For example, the scoping review by Dwyer et al. [32] identified only five empirical studies on HRO implementation across various industries, including three in healthcare, one in oil and gas, and one covering mixed industries. Veazie et al. [7] conducted a rapid evidence review in preparation for an organization-wide HRO implementation and identified only nine studies from 2010 to 2019 pertaining to HRO outcomes. Similarly, Fricke et al. [33], rapid evidence review focused on patient safety outcomes with only two included studies. While these previous reviews applied a rigorous approach to empirical evidence, the current study identified greater number of studies pertaining to HRO outcomes within the healthcare setting. Importantly, it also expands the conversation beyond patient safety to include staff safety outcomes, thereby offering a more holistic perspective on the impact of HRO principles in healthcare settings.

Most studies in this review used quantitative methods using a cross-sectional design and examined the association between HRO implementation and the incidence of safety events. Qualitative and mixed methods were used to delve into staff perspectives and perceptions concerning a particular phenomenon of HRO adoption into the healthcare organization, a process that ultimately helps comprehend the elements deemed significant by those engaged in patient care [34]. A less commonly employed design was a quasi-experimental study to determine the effect of HRO implementation on the organization's incidence of safety events. Most studies focused on the relationship between the HRO implementation and patient safety events, and qualitative approaches helped to understand the experiences of staff on HRO implementation. Opportunities remain for experimental studies to identify the effects of HRO implementation on patient safety outcomes.

The sampling included organizational entities and/or individual staff members. A large proportion of the individual staff sampled were direct care personnel, mainly nurses and physicians, which is understandable given their front-line role in the operationalization of HRO principles. Despite having direct care staff roles, other allied health personnel, such as radiology technicians, patient transporters, and rehabilitation therapy staff, were not included in the studies. Excluding other allied health professionals and non-Facility Types may overlook the impact of HRO on their practices and potential contributions to the organization's overall safety. An inclusion of varied perspectives substantiates a comprehensive understanding of the operationalization of HRO principles across different layers of healthcare organizations.

HRO research has measured HRO practices both quantitatively and qualitatively. Quantitative measurement was used in most studies to analyze the degree of the presence of HRO or the principles in the organization. Contrastingly, the implementation of HRO principles within the daily practice of perceptions was explored using qualitative approaches and often embedded within a mixed-method design. HRO principles alignment with routine actions was captured through narrative data from two mixed-methods studies and one qualitative study, with researchers analyzing these principles based on narratives that correspond with their perspectives on HRO and patient safety.

The reviewed studies examined the impact of HRO mostly on patient safety events such as medication errors and patient falls, or SSEs. Measuring the patient safety events aligns with AHRQ's recommendation of evaluating the impact of HRO implementation in improving patient safety outcomes [11]. Most of the reviewed studies consistently indicated that the implementation of HRO principles was associated with a significant reduction in patient safety events specifically with

medication errors and patient falls [29,31]. However, the relationship between HRO implementation and SSEs produced mixed results. While studies by Lyren et al. [31] and Sculli et al. [9] found a reduction in SSEs following HRO implementation, Randall et al. [19] and Sawyer et al. [28] found no significant relationship between HRO implementation and SSEs. Furthermore, Sawyer et al. study event highlighted adverse events increased with facilities that implemented HRO likely due to improved patient safety culture that enhanced reporting and detection. A key distinction lies in the study designs and methods. Lyren et al. [31] and Sculli et al. [9] assessed SSEs using pre- and post-HRO implementation data whereas Randall et al. examined the correlation between HRO maturity levels and SSEs using cross-sectional data. Sawyer et al. was the only quasi-experimental study comparing facilities that implemented HRO with those that did not, this provides a comparative perspective that is not present in the three previous studies. Both Sawyer et al. and Randall et al. examined facilities with varying levels of HRO implementation either comparing facilities with HRO implementation to those that did not adopt HRO or assessing differing maturity levels at one point in time. This variability in HRO implementation followed by the rare occurrence of SSEs, had made it difficult to observe significant effects. In contrast, prospective designs like those of Lyren et al. and Sculli et al. followed the same organization over time which provides a clearer assessment of how a consistent adoption and maturity of HRO showed a reduction of SSE. An opportunity exists for future research to explore HRO and their impact on patient safety culture, reporting behaviors, and patient safety outcomes more comprehensively.

The implementation of HRO does not only affect patient safety events but also impacts the safety mindset of the direct care staff and organizational leadership. Eight studies included in this review highlighted that the implementation of HRO significantly improved direct care staff's prioritization of patient safety, promoted the staff's psychological safety and respectful interactions, which subsequently contributed to an increased reporting of errors and improved safety climate [9,13,24–30]. The HRO promotes the direct care staff's identification of patient safety as important in their organizational operations, stresses direct care staff as drivers of operational changes, and empower staff members to promote teamwork, improve communication, learn from previous errors, and build trust within organizations [1,11]. The findings on psychological safety align with Cartland et al.'s [12] study, which addressed that staff's psychological safety helps the organization towards HRO as it promotes safety culture, adherence to the policy, and mindfulness to help staff identify and solve problems. Mindfulness promotes the principle of resilience due to continuous adaptation, which is necessary for improvements. However, the leadership perceptions on safety remains valuable, particularly those at the board and executive levels, acknowledging their essential role in fostering and sustaining an HRO culture in the organization [25,30]. Although the perception of patient safety by direct care leadership may seem inconsistent with that of executive leadership, this inconsistency can be explained by the fact that leaders of direct care staff are typically aware of the clinical realities in the organization. Future research should examine how differences in patient safety perceptions between direct care staff and leadership impact the organizational goal of achieving HRO.

Only one study examined the impact of HRO on staff outcomes. Gilmartin et al. [24] examined the relationship between HRO implementation and staff's psychological safety, employee engagement, retention rates, and overall job satisfaction, all of which demonstrated significant improvements. The review finding indicates that HRO research in healthcare organizations has focused primarily on HRO applications to improve patient safety, while little attention has paid to staff safety. In the extant body of HRO applications research, the lack of focus on staff safety in healthcare constrains the potential of HRO applications for a cultural change achieving both patient and staff safety. Given the significant correlation between staff safety and patient safety, staff's health and safety should not be secondary in the HRO application agenda. More research on the impact of HRO on staff safety is needed to motivate discussions and endorsements by different agencies and stakeholders committed to both patient and staff well-being.

## Strengths and limitations

This scoping review applied a rigorous and systematic search strategy based on an established scoping protocol. This approach allowed studies have the diversity of empirical studies spanning multiple levels in healthcare, focusing not only

on staff perceptions and patient outcomes but also on evaluating organizations as a whole and their leadership. The comprehensive scope provided insights that ranges from the patients as the recipient of care to the healthcare provider and organizational leadership. Furthermore, the review explores multidimensional aspects of healthcare by addressing both staff and patient safety, ultimately identifying a research gap in the current literature on HRO implementation and its impact on safety outcomes in healthcare.

There are limitations to consider in this scoping review. Limiting the literature search to three databases may have resulted in the exclusion of studies indexed in other databases. Also, there is a potential publication bias in the selected studies. Furthermore, the review was predisposed to language bias, as it only included studies published in English. In this scoping review, quality assessment was not conducted because the review aimed to provide a broad overview of the existing literature on HRO in relation to patient and staff safety outcomes, focusing on identifying knowledge gaps rather than critically appraising the quality of evidence.

## Conclusions

The review findings highlight the positive impact of HRO practices on patient safety. The adoption of HRO principles in healthcare demonstrated evidence for reducing adverse patient safety events and enhancing the overall culture of patient safety in healthcare. This review also disclosed a noticeable gap in HRO research regarding the effects of HRO principles on staff outcomes with only one study identified. Investigating this area could provide insights into how HROs enhance workplace safety. Research is needed to explore the impact of HRO principles comprehensively.

## Supporting information

**S1 Checklist. Preferred Reporting Items for Systematic reviews and Meta-Analyses extension for Scoping Reviews (PRISMA-ScR) Checklist.**
(DOCX)

## Acknowledgments

The authors would like to thank Bruce Smith for his valuable assistance in editorial review and proofreading this manuscript.

## Author contributions

**Conceptualization:** Michael Joshua G. Morales.

**Data curation:** Jessica Sewnath.

**Formal analysis:** Michael Joshua G. Morales, Pauline Hilton, OiSaeng Hong, Stella Bialous, Marie Martin, Jessica Sewnath, Soo-Jeong Lee.

**Funding acquisition:** Michael Joshua G. Morales.

**Investigation:** Michael Joshua G. Morales, Pauline Hilton, OiSaeng Hong, Stella Bialous, Marie Martin, Jessica Sewnath, Soo-Jeong Lee.

**Methodology:** Michael Joshua G. Morales, Pauline Hilton, OiSaeng Hong, Stella Bialous, Marie Martin, Soo-Jeong Lee.

**Project administration:** Michael Joshua G. Morales.

**Supervision:** Pauline Hilton, OiSaeng Hong, Stella Bialous, Marie Martin, Soo-Jeong Lee.

**Validation:** Michael Joshua G. Morales, Pauline Hilton, OiSaeng Hong, Stella Bialous, Soo-Jeong Lee.

**Visualization:** Michael Joshua G. Morales.

**Writing – original draft:** Michael Joshua G. Morales, Soo-Jeong Lee.

**Writing – review & editing:** Michael Joshua G. Morales, Pauline Hilton, OiSaeng Hong, Stella Bialous, Marie Martin, Jessica Sewnath, Soo-Jeong Lee.

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
