## [Decision Letter · Decision Letter 0]

15 Dec 2025

PGPH-D-25-03231

High Reliability Organizations and Healthcare Safety Outcomes on Patients and Staff: Scoping Review

Dear Dr. Morales,

Thank you for submitting your manuscript to PLOS Global Public Health. After careful consideration, we feel that it has merit but does not fully meet PLOS Global Public Health’s publication criteria as it currently stands. Therefore, we invite you to submit a revised version of the manuscript that addresses the points raised during the review process.

The manuscript has been evaluated by two reviewers, and their comments are available below.

The reviewers have raised concerns that need attention. The reviewers would like more discussion framing your study in the context of existing work. The reviewers also want further clarification on methodological aspects of your study specifically how disagreement between researchers was handled and any tools that were used for the study. The reviewers also think that the conclusion and results section can be strengthened, highlighting the results more clearly. The reviewers have provided suggestions to improve the clarity of the manuscript and suggest the manuscript be reviewed for typographical and grammar errors.

Could you please revise the manuscript to carefully address the concerns raised?

We look forward to receiving your revised manuscript.

Kind regards,

Katherine Demi Kokkinias, Ph.D.

Staff Editor

Journal Requirements:

Additional Editor Comments (if provided):

Reviewers' comments:

Reviewer's Responses to Questions

**Comments to the Author**

1. Does this manuscript meet PLOS Global Public Health’s publication criteria? Is the manuscript technically sound, and do the data support the conclusions? The manuscript must describe methodologically and ethically rigorous research with conclusions that are appropriately drawn based on the data presented.? Is the manuscript technically sound, and do the data support the conclusions? The manuscript must describe methodologically and ethically rigorous research with conclusions that are appropriately drawn based on the data presented.

Reviewer #1: Yes

Reviewer #2: Yes

2. Has the statistical analysis been performed appropriately and rigorously?

Reviewer #1: Yes

Reviewer #2: N/A

3. Have the authors made all data underlying the findings in their manuscript fully available (please refer to the Data Availability Statement at the start of the manuscript PDF file)?

The PLOS Data policy requires authors to make all data underlying the findings described in their manuscript fully available without restriction, with rare exception. The data should be provided as part of the manuscript or its supporting information, or deposited to a public repository. For example, in addition to summary statistics, the data points behind means, medians and variance measures should be available. If there are restrictions on publicly sharing data—e.g. participant privacy or use of data from a third party—those must be specified.requires authors to make all data underlying the findings described in their manuscript fully available without restriction, with rare exception. The data should be provided as part of the manuscript or its supporting information, or deposited to a public repository. For example, in addition to summary statistics, the data points behind means, medians and variance measures should be available. If there are restrictions on publicly sharing data—e.g. participant privacy or use of data from a third party—those must be specified.

Reviewer #1: Yes

Reviewer #2: Yes

4. Is the manuscript presented in an intelligible fashion and written in standard English?

Reviewer #1: Yes

Reviewer #2: Yes

Reviewer #1: This scoping review offers a timely and important synthesis of how High Reliability Organization (HRO) principles influence patient and staff safety in healthcare settings. By systematically mapping current evidence and following PRISMA-ScR standards, the paper provides clear insights into the relationship between organizational reliability, safety culture, and clinical outcomes. Its findings address a significant gap in the literature and offer practical implications for improving healthcare quality and minimizing preventable harm. The review’s rigor, relevance, and applicability make it well-suited for publication in a peer-reviewed journal.

Reviewer #2: This Scooping review seems valuable and can be publishable if authors revise minor corrections like revision in abstract, grammatical errors, content repetition, authors reasons for year ranged of scooping review etc. Please go through it and make necessary correction in the following suggestions:

1. abstract: Add the information related to health care safety outcome and its important in introduction of abstract. Method section needs to mention electronic data searches to search for evidence, criteria used data extraction method etc. Include summary of findings of the studies along with associations. Revise conclusion

2. Methods:

In search strategies, revise the last 3 line in the search strategies because these lines creates confusion in the inclusion of studies. Better to write studies beyond 2019 are also incorporated in this study as rationale

In inclusion and exclusion criteria, no need to repeat information in exclusion criteria that already mentioned in the inclusion criteria (1st 4 lines of 2nd paragraph)

Data extraction and analysis: specify the data collection tool use in the study. mention about the validation of findings disagreement between two reviewer through discussion and with third person (.....)

4. Result: in result section, first paragraph is confusing. please write clearly about the publication of studies in chronological order. You have mentioned that 8 studies are conducted in US. Please mention the setting of other 2 studies also

5. Discussion looks vague. Please revise your discussion with comparison of findings with other scooping and systematic review related to topics

6. Conclusion: Revise your conclusion with emphasis of your study findings

7. Other: Repeated information in supportive information(26-31 and 1-6), error in page number, write the citation in PRISMA Flow , Specify the reason for the studies not retrieved in the PRISMA flow diagram (figure 1)

**Do you want your identity to be public for this peer review?** For information about this choice, including consent withdrawal, please see our Privacy Policy..

Reviewer #1: **Yes:** Shazina SaeedShazina SaeedShazina SaeedShazina Saeed

Reviewer #2: No

---

## [Editor Report · Decision Letter 1]

11 Mar 2026

High Reliability Organizations and Healthcare Safety Outcomes on Patients and Staff: Scoping Review

PGPH-D-25-03231R1

Dear Dr. Morales,

We are pleased to inform you that your manuscript 'High Reliability Organizations and Healthcare Safety Outcomes on Patients and Staff: Scoping Review' has been provisionally accepted for publication in PLOS Global Public Health.

Best regards,

Julia Robinson

Executive Editor